# Retrieval of Leaf Chlorophyll Contents (LCCs) in Litchi Based on Fractional Order Derivatives and VCPA-GA-ML Algorithms

**DOI:** 10.3390/plants12030501

**Published:** 2023-01-21

**Authors:** Umut Hasan, Kai Jia, Li Wang, Chongyang Wang, Ziqi Shen, Wenjie Yu, Yishan Sun, Hao Jiang, Zhicong Zhang, Jinfeng Guo, Jingzhe Wang, Dan Li

**Affiliations:** 1Institute of Resources and Ecology, Yili Normal University, Yining 835000, China; 2College of Biological and Geographical Sciences, Yili Normal University, Yining 835000, China; 3Key Lab of Guangdong for Utilization of Remote Sensing and Geographical Information System, Guangdong Open Laboratory of Geospatial Information Technology and Application, Research Center of Guangdong Province for Engineering Technology Application of Remote Sensing Big Data, Guangzhou Institute of Geography, Guangdong Academy of Sciences, Guangzhou 510070, China; 4Guangzhou Climate and Agrometeorology Center, Guangzhou 510070, China; 5Maoming Meteorological Observatory of Guangdong Province, Maoming 525000, China; 6School of Artificial Intelligence, Shenzhen Polytechnic, Shenzhen 518055, China; 7State Key Laboratory of Resources and Environmental Information System, Institute of Geographic Sciences and Natural Resources Research, Chinese Academy of Sciences, Beijing 100101, China

**Keywords:** litchi, leaf chlorophyll content, variable selection, machine learning

## Abstract

The accurate estimation of leaf chlorophyll content (LCC) is a significant foundation in assessing litchi photosynthetic activity and possible nutrient status. Hyperspectral remote sensing data have been widely used in agricultural quantitative monitoring research for the non-destructive assessment of LCC. Variable selection approaches are crucial for analyzing high-dimensional datasets due to the high danger of overfitting, time-intensiveness, or substantial computational requirements. In this study, the performance of five machine learning regression algorithms (MLRAs) was investigated based on the hyperspectral fractional order derivative (FOD) reflection of 298 leaves together with the variable combination population analysis (VCPA)-genetic algorithm (GA) hybrid strategy in estimating the LCC of Litchi. The results showed that the correlation coefficient (*r*) between the 0.8-order derivative spectrum and LCC had the highest correlation coefficients (*r* = 0.9179, *p* < 0.01). The VCPA-GA hybrid strategy fully utilizes VCPA and GA while compensating for their limitations based on a large number of variables. Moreover, the model was developed using the selected 14 sensitive bands from 0.8-order hyperspectral reflectance data with the lowest root mean square error in prediction (RMSEP = 5.04 $\mu g\cdot {\mathrm{cm}}^{-2}$). Compared with the five MLRAs, validation results confirmed that the ridge regression (RR) algorithm derived from the 0.2 order was the most effective for estimating the LCC with the coefficient of determination (R^2^ = 0.88), mean absolute error (MAE = 3.40$\;\mu g\cdot {\mathrm{cm}}^{-2}$), root mean square error (RMSE = 4.23$\;\mu g\cdot {\mathrm{cm}}^{-2}$), and ratio of performance to inter-quartile distance (RPIQ = 3.59). This study indicates that a hybrid variable selection strategy (VCPA-GA) and MLRAs are very effective in retrieving the LCC through hyperspectral reflectance at the leaf scale. The proposed methods could further provide some scientific basis for the hyperspectral remote sensing band setting of different platforms, such as an unmanned aerial vehicle (UAV) and satellite.

## 1. Introduction

Litchi, as a typical subtropical evergreen fruit tree, is one of the important economic pillars of farmers in southern China, such as Guangdong province. The timely and rapid monitoring of the growth and nutrition of this crop is conducive to precise field management [1]. Chlorophyll absorbs sunlight and uses its energy to synthesize carbohydrates from CO_2_ and H_2_O. It plays an important role in vegetation stress, photosynthetic capacity, and physiological status and thus affects the primary production and crop harvest [2,3,4,5]. In addition, the leaf chlorophyll content (LCC) is closely related to the nitrogen (N) content [6,7] and can be used as a close proxy for the N concentration at the leaf level [8,9]. The nutritional status of crops is also closely related with the chlorophyll. The laboratory chemical measurement of LCC is destructive and relatively time- and labor-consuming. It is difficult to meet the practical demands of precise crop management in large or regional fields [10]. Thus, it is crucial to create quick, non-destructive, and efficient techniques that can deliver precise LCC estimations.

With the advancement of remote sensing techniques, hyperspectral remote sensing data, with their abundance of data, continuity, and rich hidden characteristics, have been widely used to non-destructively and accurately monitor crop chlorophyll contents [9,11]. However, there is a significant chance of over-fitting when modeling spectral data with a large number of wavelength variables and relatively few samples, which will lead to subpar or ineffective prediction results of multivariable estimation models. Therefore, efficient variable (feature) selection techniques have taken center stage in the analysis of hyperspectral remote sensing data. By alleviating the dimensionality curse, variable selection can yield faster and more cost-effective variables, improve the predictive performance of the chosen variables, and make it easier to understand and justify the models that are generated [12]. Yun et al. [13] confirmed the importance and necessity of variable selection in complex analysis systems. In recent decades, more and more experts and scholars have invested in relevant research and proposed many variable selection algorithms. These variable selection algorithms can be summarized into four types: (1) wavelength point-based selection algorithm, which is characterized by taking each wavelength variable as a unit and studying it based on four factors: different variable initialization, modeling methods, evaluation indicators, and selection strategies, and finally selecting the best combination of variables, such as the successive projections algorithm (SPA) [14], Monte Carlo uninformative variables elimination (MC-UVE) [15], competitive adaptive reweighted sampling (CARS) [16], and variable combination population analysis (VCPA) [17]; (2) wavelength range selection algorithm; its characteristic is that each wavelength range is taken as a unit, and then, the best combination of interval variables is selected according to different search strategies. Each interval is composed of several continuous variables, which is consistent with the continuous band characteristics of vibration and rotation spectra, making the modeling more interpretable, such as interval partial least-squares (iPLS) [18], interval random frog (iRF) [19], fisher optimal subspace shrinkage (FOSS) [20], and the interval variable iterative space shrinkage approach (iVISSA) [21]; (3) hybrid variable selection algorithm; its characteristic is to combine two or three existing algorithms and optimize the algorithm by combining the advantages of the algorithm, such as CARS-SPA [22] and iPLS-SPA [23]; (4) improved variable selection algorithm, which is based on the method of improving at least one of the four factors of the variable initialization, model method, evaluation index, and selection strategy, such as stability competitive adaptive reweighted sampling (SCARS) [24] and variable permutation population analysis (VPPA) [25].

Leaf reflectance is an efficient method for determining the LCC [26,27,28] since an increase or reduction in LCC may produce more or less absorption in blue and red wavelengths, which ultimately alters the spectral reflectance of leaves. In recent years, hyperspectral reflectance data have been used in some studies to estimate LCC at various scales based on the reaction of leaf reflectance to LCC (Table 1). The current research on crop LCC is essentially concerned with analyzing the difference in LCC inversion from two levels of the spatial scale effect and wide and narrow band spectral resolution. The remote sensing data acquisition platforms are constantly updated from aerospace and aviation to low altitude; LCC inversion models are continuously improved from traditional empirical models, such as linear regression (LR), to physical models, such as PROSPECT, and then to hybrid inversion models by using machine learning algorithms (MLAs). However, in the studies mentioned above, hyperspectral data only use original spectral reflectance or mathematical transformation forms, such as first and second derivatives, and ignore the potential information contained between them, which may result in the loss of crucial information and a decline in model accuracy. Zhang et al. [29] analyzed the correlation between hyperspectral reflectance through fractional order derivatives (FODs) and heavy metal content in maize leaves and found that FODs can expand the selection space of sensitive bands. Moreover, few studies have considered the potential interaction impact of variables through random combinations, while the majority of studies use a single variable selection approach.

Hence, to address the above difficulties, this study proposed machine learning regression algorithms (MLRAs) using hyperspectral reflectance data for litchi LCC estimation. The following are the main objectives of this study: (1) to explore the impact of FODs on litchi leaf spectra and comparatively analyze the correlation between the litchi LCC and FOD spectra based on Pearson’s correlation; (2) to explore the hybrid variable selection algorithm, VCPA coupled with the genetic algorithm (GA), and its potential application in retrieving the LCC of litchi; (3) to develop MLRAs and evaluate the accuracy of the optimal litchi LCC estimation model based on FOD-VCPA-GA. 

## 2. Results

### 2.1. Correlation Analysis between LCC and FOD Spectra

Figure 1a displays the leaf spectral curves of litchi with various LCCs. As shown in this figure, the reflectance curves of litchi leaves with different LCCs included one reflection peak (about 550 nm) and two absorption valleys (450 nm and 670 nm) in the 400–780 nm (visible) range. Chlorophyll, which has a strong absorption of blue and red light and a high reflection of green light, is primarily responsible for this property [39]. The leaf reflectance gradually dropped in the vicinity of 550 nm as the LCC increased.

In the range of 670–750 nm, there was a reflection “steep slope”, and as the LCC increased, the reflection curve of litchi leaves shifted to the long wave direction. After 750 nm, there were no overt variations in the leaf reflectance of litchi with various LCCs. At 1450 nm and 1950 nm, there were two absorption valleys that were mostly brought on by the effect of leaf water content. The spectral features of litchi leaves described above were comparable to those of green plant leaves.

The linearity of the link between two variables can be confirmed via correlation analysis. We can determine the existence of a linear relationship between two variables, its strength, and whether it is positive or negative by looking at the correlation coefficient (r). In this study, Pearson’s correlation coefficients for LCC and FOD spectra (0–2 order) were calculated and tested at the 0.01 significance level (*r* > 0.1465). A thorough outcome was plotted in Figure 1b. The position of the band with a positive and negative association with LCC fluctuated with the continual increase in order, and it was primarily dispersed in the visible near-infrared (VIS-NIR) range (400–900 nm). The reflectance in the 400–497 nm, 665–679 nm, and 756–900 nm regions was positively correlated with the LCC for the original spectral (0 order) data, while the reflectance in the 498–664 nm and 680–755 nm ranges was negatively correlated. The maximum absolute value of the correlation coefficient was shown at 709 nm (*r* = −0.8542).

Table 2 displays the statistics for the number of bands that passed the 0.01 significance test (0–2 order). As shown in Table 2, the overall number of bands passing the 0.01 significance test and the number of bands positively connected to the LCC were reduced as the order increased, while the number of bands negatively related to the LCC essentially increased first and then decreased. At 756 nm of the 0.8 order, the correlation coefficient reached its greatest value (*r* = 0.9179), followed by 720 nm of the 1.8 order (*r* = 0.9020) and 723 nm of the 1.6 order (*r* = 0.9018). These bands all appeared in the red-edge region, which is an important indicator area for describing the state of plant pigments. In conclusion, the results of correlation analysis showed that the correlation between FOD spectra and the LCC of litchi was greater than the commonly used first- and second-order derivatives, and it is worthwhile to further investigate its potential for estimating LCC.

### 2.2. Performance of VCPA-GA Hybrid Strategy for Variable Selection

A VCPA-GA hybrid strategy was proposed to further optimize and extract sensitive band information from the spectra of 400–900 nm. Figure 2 shows the distribution of sensitive bands screened using the VCPA-GA hybrid strategy. Variable selection is a critical and necessary step for the LCC estimation models, as illustrated in Figure 2, where the variable regions selected using VCPA-GA are similar but the number of sensitive bands selected has been greatly reduced, with the majority of them being concentrated around 590 nm, 760 nm, and 840 nm. The spectral reflectance near 590 nm and 760 nm was strongly related to the LCC, which was basically consistent with the results of the Pearson correlation analysis.

Table 3 shows the statistical results of the VCPA-GA hybrid strategy based on the 0–2-order dataset, including the number of selected variables (N_var_), the number of optimal PLS latent variables (N_lvs_), the root mean square error in calibration (RMSEC),the root mean square error in cross validation (RMSECV), and the root mean square error in prediction (RMSEP). As seen in Table 3, the number of chosen sensitive bands did not exhibit any clear regularity as the order increased. The 0.2 derivative was the most frequently chosen order among them (N_var_ = 54), while the original spectrum had the fewest bands (N_var_ = 5). The prediction performance of the 0.8 order (RMSEP = 5.04) was better than that of the other orders, followed by that of the 1.4 order (RMSEP = 5.24) and 1.8 order (RMSEP = 5.25). FOD spectrum has some potential in determining the LCC of litchi. The variable selection is a crucial and necessary step in FOD spectral data mining. VCPA-GA hybrid strategy may fully exploit the benefits of the VCPA and GA algorithms and comprises a great enhancement to the FOD spectral variable selection.

### 2.3. MLRAs for Estimating the LCC of Litchi

After selecting the best sensitive band combination of the 0–2-order derivative through the VCPA-GA hybrid strategy, five machine learning regression models were constructed for estimating the LCC of litchi. The training, testing, and validation results of MLRAs are shown in Table 4. For the training set, the XGBoost model performed best for all datasets of the 0–2 order, with R^2^ reaching 0.99, followed by the RF (R^2^: 0.85~0.92) and SVR (R^2^: 0.83~0.88) models. Among them, the training effect for the 0.2-order derivative data with the XGBoost model was the best with the lowest MAE and RMSE value (MAE = 1.21 μg·cm^−2^, RMSE = 1.70 μg·cm^−2^), followed by that of the 0.4 order with XGBoost (MAE = 2.06 μg·cm^−2^, RMSE = 2.75 μg·cm^−2^) and the 1.6 order with RF (MAE = 2.42 μg·cm^−2^, RMSE = 3.19 μg·cm^−2^). There was no glaring rule discovered for the testing set. The MAE values of SVR and GPR were typically high in all models of 0–2-order spectra datasets, and the testing effect of the RR model of the 1.8 order was the best (R^2^ = 0.85, MAE = 3.59 μg·cm^−2^, RMSE = 4.67 μg·cm^−2^).

The validation of the MLRAs for predicting the LCC was conducted using an independent dataset (n = 47). The validation performance varied between orders and models, just as the training and testing sets did, and it remained largely steady at the 0.2 order in five MLRAs in terms of R^2^, MAE, RMSE, and RPIQ. The rankings were as follows: RR (R^2^ = 0.88, MAE = 3.40 μg·cm^−2^, RMSE = 4.23 μg·cm^−2^, RPIQ = 3.59) > GPR (R^2^ = 0.88, MAE = 3.55 μg·cm^−2^, RMSE = 4.29 μg·cm^−2^, RPIQ = 3.86) > XGBoost (R^2^ = 0.85, MAE = 3.90 μg·cm^−2^, RMSE = 4.84 μg·cm^−2^, RPIQ = 2.67) > SVR (R^2^ = 0.81, MAE = 3.94 μg·cm^−2^, RMSE = 5.37 μg·cm^−2^, RPIQ = 3.09) > RF (R^2^ = 0.80, MAE = 4.28 μg·cm^−2^, RMSE = 5.47 μg·cm^−2^, RPIQ = 2.57). Our results indicated that the accuracy of the LCC assessment of litchi was somewhat enhanced by the FOD spectrum and MLRAs, and especially RR, GPR, and XGBoost, can predict the LCC of litchi well in the two study areas.

The scatterplots of measured and estimated LCCs based on the best MLRA at 0–2 orders are illustrated in Figure 3a–k. The figure illustrates that the sample data for the best estimation models at the 0–2 order were almost evenly distributed near the 1:1 line, indicating no apparent overestimation or underestimation. The models based on the 0-GPR, 0.2-RR, 0.6-GPR, 0.8-RR, and 1-RR all had RPIQ values above 3.0, further demonstrating the feasibility and effectiveness of using the FOD spectra to predict the LCC of litchi.

## 3. Materials and Methods

### 3.1. Study Area

Guangdong is the most important litchi-producing area in China, with the cultivation area and output ranking first among all provinces and regions in the country. In this study, two commercial ‘Guiwei’ litchi orchards, normally operated by local farmers, were selected as the study area (Figure 4). One (Litchi orchard 1) was located in Yangxi County of Yangjiang City (111°22′–111°48′ E, 21°29′–21°55′ N), and the other (Litchi orchard 2) was in Dianbai District of Maoming City (110°54′–111°29′ E, 21°22′–21°59′ N). The above two areas belong to a subtropical monsoon climate, with sufficient sunshine, abundant rainfall, and a pleasant climate. The annual average temperature is about 23 °C, the vegetation is evergreen, and the flowers are always in bloom. Litchi is one of the specialties of the two places. Data collection was carried out at the flower bud differentiation (28 December 2020) and the blooming florescence (19 March 2021) stages. The selected trees were in good condition.

### 3.2. Hyperspectral Measurements and Preprocessing

In total, 49 ‘Guiwei’ litchi trees (25 in Yangxi county and 24 in Dianbai District) were selected. Moreover, the longitude and latitude information of each tree was recorded using a GPS. Six leaves of each litchi tree were collected and put into fresh-keeping bags for later spectral measurements and chlorophyll extraction. Hyperspectral data for the litchi leaves were measured using an ASD FieldSpec3 spectrometer (Analytical Spectral Devices, Inc., Boulder, CO, USA) [5] with the range 350–2500 nm. To reduce the influence of the solar altitude angle, the spectral measurement was carried out at 10:00–14:00 Beijing time with cloudless and sunny weather. Every 3–5 min, the spectral reflectance was calibrated using a standardized whiteboard (25 cm × 25 cm, 100% reflectance). Ten spectral curves were collected for each leaf sample, with a measurement interval of 0.1 s. The average value of the 10 spectral curves was taken as the spectral data of this leaf sample. In total, 294 leaves were collected. There were 294 sets of data. One group of data was removed because of data damage. Thus, 293 sets of data were used for the analysis.

The edge bands 350–399 nm and 2401–2500 nm with high optical noise were removed [40]. The remaining spectral curves, as the original reflectance spectrum, were smoothed using the Savitzky-Golay filtering method [41]. Then, the fractional order derivative (FOD) of the smoothed spectral data was calculated with the Grünwald-Letnikov (G-L) algorithm as shown in the Equation (1) [42] using a program in Matlab R2021a (The MathWorks Inc.: Natick, MA, USA).
(1)$$\frac{d^{v}f\left(x\right)}{dx^{v}}\approx f\left(x\right)+\left(-v\right)f\left(x-1\right)+\frac{\left(-v\right)\left(-v+1\right)}{2}f\left(x-2\right)+\cdots \frac{\Gamma \left(-v+1\right)}{m!\Gamma \left(-v+m+1\right)}f\left(x-m\right)$$
where Γ is the Gamma function, ***x*** is the value of the corresponding point, ***m*** is the difference between the upper and lower bounds of the differential, and v is the order allowed to vary from 0–2 (increment by 0.2 at each step) in this study. In addition, v=0 indicated that the spectral data comprised the original reflectance.

### 3.3. Determination of the LCC

In this study, SPAD-502 plus portable chlorophyll meter (minola Osaka company) was used to measure the leaf chlorophyll content of litchi. Since the value read from the SPAD-502 plus is unitless, it needs to be converted into LCC (μg·cm^−2^), and the conversion process was completed using Equation (2) [43].

The chlorophyll content of the selected trees ranged from 12.44 to 73.95 μg·cm^−2^. The descriptive statistics of leaf chlorophyll content are presented in Figure 5.
(2)$$C_{ab}=6.34299\times \exp\left(SPAD\times 0.04379\right)-6.10629\;\left(RMSD=5.4\;\mu g\cdot {\mathrm{cm}}^{-2}\right)$$

### 3.4. VCPA-GA Hybrid Strategy for Variable Selection

VCPA is a relatively new variable selection algorithm. The first step is to use an exponentially decreasing function (EDF) to count the remaining variables. Binary matrix sampling (BMS) [44] is utilized in each EDF run to create the population of various variable combinations. Then, using the model population analysis (MPA) [45], the variable subset with the lowest cross validation root mean square error (RMSECV) was found using the top 10% of the sub models. When all EDF runs are finished, VCPA looks through the 14 remaining variables to get the best variable subset. GA uses the selection, exchange, and mutation operators to describe the biological world’s natural selection and genetic mechanisms. Through continuous genetic iterations, the variables with better objective function values are retained, and the variables with lower objective function values are deleted until the desired results are obtained. This has been widely used in feature variable screening [46].

The two main steps of the VCPA-GA hybrid method are shown in Figure 6. This strategy’s specifics was described in Yun et al. [47]. A calibration set (193 samples) and an independent test set (100 samples) were created from the dataset. Once the model establishment and variable selection were completed in the calibration set, an independent test set was used to verify the calibration model. As a modeling technique, partial least square (PLS) was employed. Using 5-fold cross validation (CV) with a range of 1 to 10, the ideal number of PLS latent variables was determined. All data were centered before preprocessing so that the mean of each column would be zero. Fifty replications of VCPA-GA (ɷ = 100, ɷ is the number of variables left for GA) were performed in order to assess the model’s repeatability and produce statistical results. All calculations were implemented using MATLAB (Version 2021a, the MathWorks, Inc) on a desktop computer equipped with an 12th Gen Intel(R) Core (TM) i9-12900H 2.50 GHz CPU and 32GB of RAM memory, and the operating system was Windows 11.

### 3.5. The Evaluation of the Proposed MLRMs

For this study, hyperspectral sensitive bands selected using a VCPA-GA hybrid strategy were taken as independent variables with LCCs as dependent variables. Then, 293 measured LCC values were randomly divided into three parts: 187 as a training set, 59 as a testing set and 47 as a validation set for validating model performance, as shown in Figure 5.

Five MLRAs were selected to explore and analyze hyperspectral reflection data for LCC modeling based on their fast training, strong performance, and popularity in different application fields. These five MLRAs were Ridge regression (RR), random forest (RF), extreme Gradient Boosting (XGBoost), support vector regression (SVR), and Gaussian processes regression (GPR). Here, RR [48] is a biased estimation regression method specially used for the analysis of collinear data. It is essentially an enhanced least squares estimate technique. It is more practical and dependable to derive regression coefficients by giving up the least square method’s impartial aspect, but at the expense of losing some information and lowering accuracy. As for the RF model [49], decision trees are built for each sample that is extracted based on RF using the bootstrap resampling approach, and the predicted average values of all the decision trees are used as the final prediction results. A distributed gradient enhancement toolkit called XGBoost [50] has been tuned for great performance, adaptability, and portability. It provides a decision tree with gradient boosting (GBDT). Being more than ten times faster than standard toolkits, it is now the best and quickest open source improvement tree toolkit. Prior to moving on to linear modeling, SVR [35] maps training samples to a high-dimensional space and then transforms a nonlinear problem in a low-dimensional space into a linear problem in a high-dimensional environment. Here, nonlinear issues were converted into linear ones using a radial basis function. GPR [51] is a nonparametric model for regression analysis of data using Gaussian process priors. It is based on the Bayesian framework. By using past data for training, it can convert a prior distribution into a posterior model and produce predictions with statistical significance. The above five MLRAs were implemented using the scikit learn Python package.

The agreement between the measured and predicted LCC values was evaluated using the coefficient of determination (R^2^), mean absolute error (MAE), root mean square error (RMSE), and ratio of performance to inter quartile distance (RPIQ) generated during prediction (Equations (3)–(6)).
(3)$$R^{2}=1-\frac{\sum_{i=1}^{n}{\left(y_{i}-\hat{y_{i}}\right)}^{2}}{\sum_{i=1}^{n}{\left(y_{i}-\bar{y_{i}}\right)}^{2}}$$
(4)$$\mathrm{MAE}=\frac{1}{n}\sum_{i=1}^{n}\left|\left(y_{i}-{\hat{y}}_{i}\right)\right|$$
(5)$$\mathrm{RMSE}=\sqrt{\frac{\sum_{i=1}^{n}{\left(y_{i}-\hat{y_{i}}\right)}^{2}}{n}}$$
(6)$$\mathrm{RPIQ}=\frac{Q_{3}-Q_{1}}{RMSE}$$
where *n* is the number of samples, $y_{i}$ is the ith measured LCC of each sample, $\hat{y_{i}}$ is the *i*th estimated LCC of each sample, $\bar{y_{i}}$ is the mean LCC, and *Q*1 and *Q*3 are the first and third quartiles, respectively.

## 4. Discussion

The LCC is a key indicator of a crop’s physiological status, and changes in it can be used to assess a crop’s photosynthetic ability, growth and development stage, nutrition, stress from humans or the environment, illnesses, and pests. Hyperspectral remote sensing technology has become a non-destructive way to estimate the LCC and may provide detailed information about how vegetation differs from soil, water, and other ground objects in terms of its spectral reflection characteristics. Numerous spectral transmission techniques have been studied in the past, such as integer derivatives, continuum-removal transformations, and mathematical transformations. Integer derivatives are particularly good at enhancing absorption features, lowering background noise, and eliminating baseline drafts [52]. However, they cannot detect gradual tilts or curvatures and useful target variables. In recent times, FOD has received an increasing amount of attention in the processing of hyperspectral data to widen the selection space for sensitive bands. In this study, we calculated the 0–2-order derivative of spectral reflectance of litchi leaves in increments of 0.2. Pearson correlation analysis showed that the absolute value of the correlation coefficient between the 0.8-order derivative spectrum at 756 nm and LCC reached a maximum, with the *r* of 0.9179 (Table 2). The proposed VCPA-GA hybrid strategy had the best performance in the FOD datasets. Especially, the generalization of the proposed hybrid variable selection strategy had RMSEP values of 5.04, 5.24, and 5.25 $\mu g\cdot {\mathrm{cm}}^{-2}$ for the LCC using 0.8-, 1.4-, and 1.8-order spectral data, respectively (Table 4). Compared with that of the first and second-order derivatives, the accuracy of the LCC estimation model based on the FOD was significantly improved. An explanation for this may be because compared to integer-order spectral data, the FOD spectra offer a superior balance among spectral resolution, spectral information, and noise.

The findings of our research are consistent with the previous research conclusions to a certain extent. Cui et al. [53] investigated the potential of using the FOD for estimating the soil copper content and found that the model using the 0.8-order FOD spectra performed the best, and the R^2^ and RPD of the validation set were 0.6416 and 1.63, respectively. Jin and Wang [54] created hyperspectral indices using FOD spectra to retrieve the leaf mass per area (LMA), and results showed that the 0.3-order FOD indices provided the highest accuracies to trace LMA and at the same time had the least sensitivity to random noise. In short, the FOD spectra are, in general, superior or at least compatible to the original reflectance or first- and second-order derivatives and could further promote the practical application of hyperspectral remote sensing in estimating plant physiological and biochemical parameters, as mentioned above. Thus, we suggest that FOD analysis is efficient to identify the best band combination that could be applied to a large measurement database with a wide variety of plant leaves and field conditions from various remote sensing platforms.

Variable selection technology plays a key role in eliminating irrelevant or uninformative variables and reducing data dimension in hyperspectral data. Yun et al. [47] used the VCPA-based hybrid strategy with iteratively retaining informative variables (IRIVs) and GA to select the optimized variables in near-infrared (NIR) spectral datasets for beer, cotton, and tablets. The findings demonstrated that when compared to other approaches, the VCPA-IRIV and VCPA-GA significantly improve model prediction performance and that the modified VCPA step is a very successful method for removing the unhelpful variables. This also provides methodological support for our study. In this study, VCPA gradually reduced the number of variables based on EDF until all hyperspectral bands were reduced and optimized. Then, a modified version of VCPA was combined with GA to create a hybrid approach for variable selection in order to get beyond the current limiting problem associated with GA for a high number of variables. By choosing too few variables, VCPA has another problem that our hybrid strategy can assist in overcoming. The original VCPA only chooses less than 14 variables, but it has components that could cause the variable space to continuously contract. Although GA is a useful optimization tool, it has a number of limitations when working with many variables. There were 501 variables in this litchi hyperspectral dataset from 400 nm to 900 nm. Finding the ideal variable subset for GA would be exceedingly challenging given this enormous variable space. The variable space decreased from 501 to 100 when modified VCPA was used as the initial step, making it much simpler to identify the ideal variable subset in this highly compressed and optimal space. It is clear from Table 3 that the RMSEC and RMSECV decrease as the order increases, indicating that the variable space is constantly optimized. Additionally, the 0.2-order derivative sensitive band combination chosen by VCPA-GA for LCC prediction using the RR model has the best accuracy. Compared with previous studies, our research proved that the suggested VCPA-GA hybrid approach may successfully be applied to hyperspectral reflectance with FODs. It could also ensure MLRA’s accuracy and avoid model overfitting.

MLRAs, such as SVR, RF, BPNN, and kernel-based extreme learning machine (KELM), have been widely used for estimating crop biochemical properties [32,33,36,55]. In our study, for the purpose of investigating and evaluating FOD spectral data optimized using the VCPA-GA approach for litchi LCC modeling, five MLRAs were developed taking into account their quick training, good performance, and popularity in numerous application areas. A comparison of them revealed that the accuracy of the models was different for the data of various FOD spectra. Among them, the RR model, based on 0.2-order derivative spectra, can estimate the LCC of litchi well. The performance of GPR and XGBoost closely followed the performance of RR (Table 4) in terms of R^2^, RMSE, and RPIQ. The stochastic gradient of XGBoost, which enhances the method, may prevent overfitting, can enhance prediction accuracy, and can be used to explain why it has greater accuracy. Additionally, the XGBoost ensemble can handle noisy data based on the deployment of a number of decision-based tree classifiers. There are numerous such instances where the XGBoost model was effectively used to forecast soil characteristics and nutrients [56,57]. Future research could also look into combining radiative transfer models (RTMs) and machine learning algorithms to accurately estimate the chlorophyll content at both the leaf and canopy scales, in addition to investigating other advanced machine learning techniques, such as stochastic gradient boosting (SGB), Cubist (CB), and deep learning.

## 5. Conclusions

In this study, we investigated the performance of five MLRAs and assessed the potential of fractional order derivatives and a VCPA-GA hybrid variable selection strategy to enhance the hyperspectral estimate of litchi LCC. Compared with the common first and second derivatives, the correlation coefficient between the FOD spectrum and LCC was improved, reaching 0.9179 at the 0.8 order (756 nm), followed by the 1.8 order (0.9020, 720 nm) and 1.6 order (0.9018, 723 nm). The VCPA-GA hybrid method improved upon VCPA’s ability to shrink the variable space constantly, and combined it with GA for further optimization. To investigate how this hybrid approach could be improved, hyperspectral datasets (0–2 order) of litchi leaves were used. The findings demonstrated that the VCPA-GA hybrid strategy fully utilizes the benefits of both VCPA and GA while compensating for their shortcomings. It fixes the issue of VCPA’s propensity to choose fewer variables and removes GA’s restrictions when working with a large number of variables. Additionally, as compared to the commonly used first- and second-order derivatives, this hybrid strategy performs noticeably better with FOD spectral data, demonstrating the effectiveness of employing FOD spectral data to compress and optimize the variable space. As a result, for FOD spectral data, VCPA-GA is an effective substitute for variable selection approaches.

From the performance of the MLRAs, we found that the training effect of the XGBoost algorithm was the best for the 0 order, with the highest R^2^ (0.99) and lowest MAE (0.53$\;\mu g\cdot {\mathrm{cm}}^{-2}$) and RMSE (0.71$\;\mu g\cdot {\mathrm{cm}}^{-2}$). During validation, RR also showed the highest accuracy at the 0.2 order, with R^2^ = 0.88, MAE = 3.40$\;\mu g\cdot {\mathrm{cm}}^{-2}$, RMSE = 4.23$\;\mu g\cdot {\mathrm{cm}}^{-2}$, and RPIQ = 3.59. It is important to note that the VCPA-GA hybrid method is a broad one that may be used with other optimization or variable selection strategies to obtain even greater optimization. Although it was used in this study based on hyperspectral datasets of litchi leaves, it might also be used with other high-dimensional datasets from scales including the canopy, landscape, and region.

## Figures and Tables

**Figure 1 plants-12-00501-f001:**
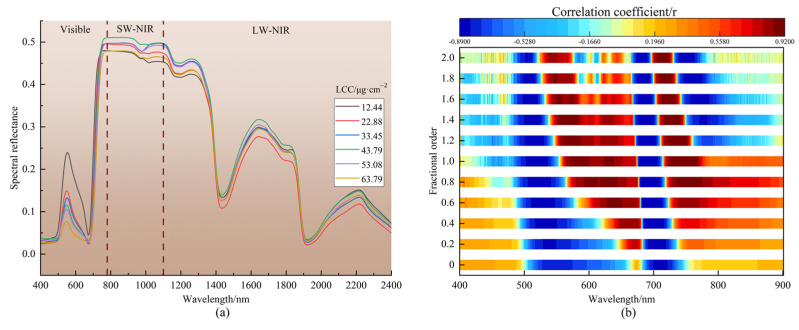
Leaf spectral curves of litchi with different LCCs (**a**); correlation coefficients between LCC and hyperspectral reflectance between 400 and 900 nm (0–2 order, 0.2 per step) (**b**). 0 order refers to the original reflectance, the dash line refers to the cutting line of the different spectral regions.

**Figure 2 plants-12-00501-f002:**
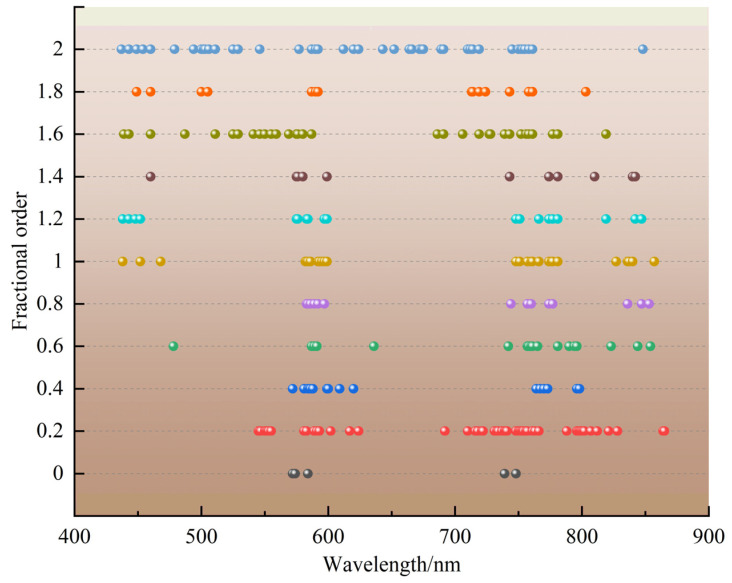
Distribution of hyperspectral sensitive bands with the VCPA-GA hybrid strategy (0–2 order), Dots of the same color respectively represent the characteristic variables screened out under different fractional order processing.

**Figure 3 plants-12-00501-f003:**
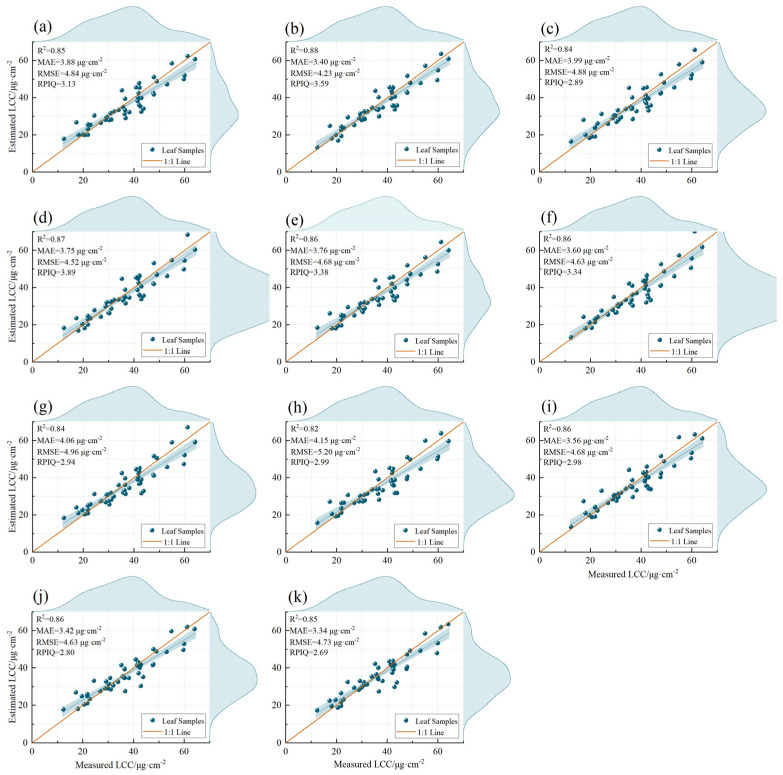
Scatterplots with the marginal histograms of measured and estimated LCCs based on the best MLRA at the 0–2 order. (**a**): 0-GPR; (**b**): 0.2-RR; (**c**): 0.4-GPR; (**d**): 0.6-GPR; (**e**): 0.8-RR; (**f**): 1-RR; (**g**): 1.2-SVR; (**h**): 1.4-RR; (**i**): 1.6-RR; (**j**): 1.8-RF; (**k**): 2-XGBoost.

**Figure 4 plants-12-00501-f004:**
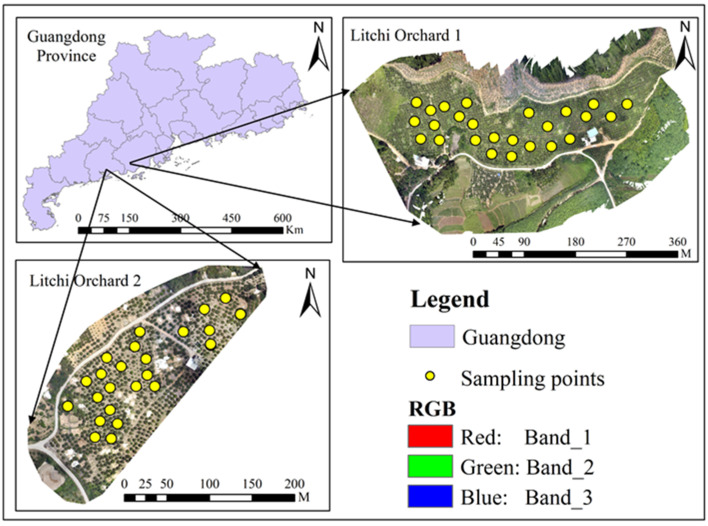
Distribution of sampling sites in Guangdong Province of China. Litchi orchard 1: Yangxi County, Yangjiang City; Litchi orchard 2: Dianbai District, Maoming City.

**Figure 5 plants-12-00501-f005:**
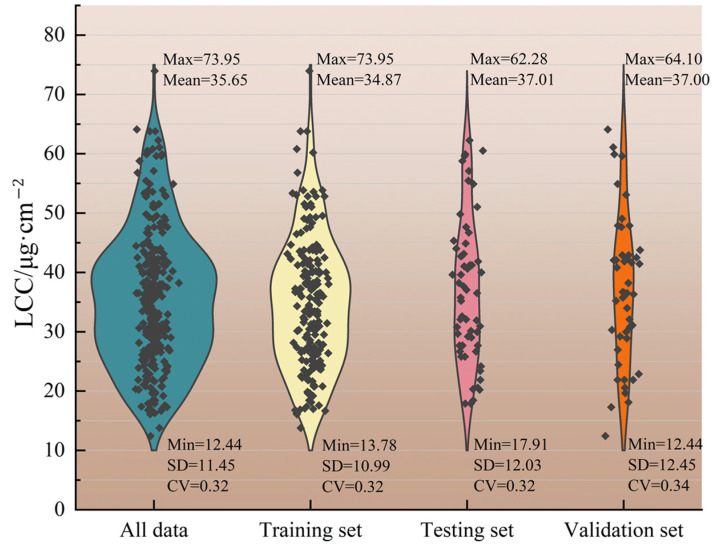
Statistical results of the litchi LCC for the training, testing, and validation datasets (SD: standard deviation, CV: coefficient of variation), the little dark gray dots in the diamond shape are the samples.

**Figure 6 plants-12-00501-f006:**
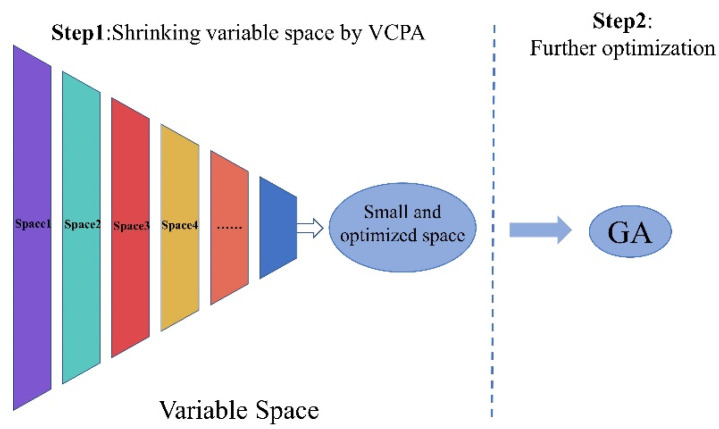
Two main steps of the VCPA-GA hybrid strategy.

**Table 1 plants-12-00501-t001:** Short overview of LCC monitoring through remote sensing.

Data Source	Name of the Sensor	Type of Spectra or Image Data	Methods	Study Area	Object of Study	Regression Statistics	Research Contents	Reference	Year of References
Airborne	Compact Airborne Spectrographic Imager (CASI)	Hyperspectral remote sensing imagery (HIS)	Lookup-table (LUT)-based inversion	Ten black spruce stands near Sudbury, Ontario	Ten black spruce	R^2^ = 0.47, RMSE = 4.34 μg/cm^2^	Estimated LCC from the CASI imagery by combining the geometrical-optical model 4-Scale and the modified leaf optical model PROSPECT	[30]	2008
Ground-based	ASD FieldSpec spectrometer	Hyperspectral remote sensing reflectance data	Narrowband vegetation indices (VIs)	The Naeba Mountains, Japan	Beech leaves	CI (R^2^ =0.73, WAIC = 2241.5, RPD = 1.76) D2 (R^2^ = 0.71, WAIC = 582.4, RPD = 1.94)	Evaluated the performances of hyperspectral indices for both leaf types within beech canopies, developed a new index for estimating LCC in both sunlit and sun-shaded areas.	[5]	2017
Ground-based	Gaia hyperspectral imaging system	HSI	Linear extrapolation method	The experimental greenhouse of China Agricultural University	Potato	R^2^ = 0.8682	Inverted the LCC of potato by using the selected optimal red edge position	[31]	2018
Ground-based	Hyperspectral lidar (HSL) system	HSL	PROSPECT-4 model, support vector regression (SVR)	Junchuan County, Suizhou, China	Rice	PROSPECT-4 model inversion (R^2^ = 0.55)	Investigated the possibility of estimating foliar Chl through the PROSPECT-4 model using the HSL system.	[32]	2018
Unmanned Aerial Vehicle (UAV)	Parrot sequoia multi-spectral sensor	Multi-spectral images (MSIs)	Machine learning regression algorithms (MLRAs)	ICAR research complex for NEH region at Umiam, Meghalaya	Maize	R^2^ = 0.904, RMSE = 0.057 mg/gm	Estimated the LCC of a standing maize plant from multi-spectral UAV images by using machine learning algorithms.	[33]	2019
UAV	Cubert S185 hyperspectral sensor	HSI	LR (linear regression), SVR (support vector regression)	Luozhuang village, Zhangziying Town, Daxing District, Beijing, China	Soybean and maize	MCARI1 for soybean (MAE = 1.617) MCARI/OSAVI for maize (MAE = 2.422);	Retrieved canopy SPAD values of maize and soybean by using the 16 VIs at different observation angles and their combinations.	[34]	2020
Satellite	Landsat-8 Operational Land Imager (OLI)	MSI	VIs (vegetation indices), MLRAs (machine learning regression algorithms), LUT (lookup-table)-based inversion, and hybrid regression approaches	Shunyi District, Beijing, China	Winter wheat	MTVI2 (RMSE = 5.99 µg/cm^2^, RRMSE = 10.49%) GPR (RMSE = 5.50 µg/cm^2^, RRMSE = 9.62%) LUT (RMSE = 8.08 µg/cm^2^, RRMSE = 14.14%) AL-GPR (RMSE = 12.43 µg/cm^2^, RRMSE = 21.77%)	Evaluated capabilities and potentials of Landsat-8 (OLI) imagery using four different retrieval methods for LCC modeling	[35]	2020
UAV	Cubert S185 hyperspectral sensor	HSI	MLR (multi-variable linear regression), RF (random forest), BPNN (backpropagation neural network), and SVM (support vector machine)	Yucheng Comprehensive Experiment Station (YCES) of the Chinese Academy of Sciences	Maize and wheat	SVM for maize (R^2^ = 0.83, RMSE = 5.80, MRE = 0.12); SVM for wheat (R^2^ = 0.78, RMSE = 2.80, MRE = 0.11)	Examined the effects of spectral information and spatial scale of unmanned drone images, as well as phenological types and phenology, on LCC estimation of maize and wheat.	[36]	2020
Satellite	Sentinel-2	MSI	PROSPECT-5 leaf optical model	The Borden Forest Research Station	Mixed temperate forest	R^2^ = 0.849, RMSE = 0.304 μg/cm^2^	Estimated LCC from Sentinel-2 (MSI) data via a physically based, two-step inversion approach	[37]	2021
UAV	Pika L hyperspectral imaging system	HSI	Narrowband vegetation indices (VIs) Multiple linear regression (MLR)	A commercial wine estate at the eastern base of Helan Mountain in Ningxia Province, China	Wine grapes	(D735 − D573)/(D735 + D573) (R^2^ = 0.50)	Investigated the SPAD changes of grape leaves at different growth stages, and explored a new method for predicting these parameters using hyperspectral imaging.	[38]	2021
Ground-based	ASD FieldSpec spectrometer	Hyperspectral reflectance data	VIs, PROSPECT, PLSR, SVR	Changchun, China	Different plant species (trees, bushes, and lianas)	Modified difference ratio index (MDRI), R^2^ = 0.92, RMSE = 5.65 μg/cm^2^	Developed a new algorithm for estimating the LCC of different plant species by combining SIs (spectral indices) with multi-angular hyperspectral reflectance of leaves.	[10]	2022

**Table 2 plants-12-00501-t002:** Statistical table of the number of spectral bands passing the 0.01 significance test (0–2 order).

Orders	Tb	Pb	Nb	rmax	Corresponding Bands/nm
0	1946	1720	226	0.8542	709
0.2	1953	1756	197	0.8722	704
0.4	1778	1609	169	0.8835	698
0.6	1631	1426	205	0.8884	694
0.8	1535	1235	300	0.9179	756
1	1340	931	409	0.8929	756
1.2	1264	515	649	0.9015	742
1.4	1086	406	680	0.8994	726
1.6	963	375	588	0.9018	723
1.8	701	290	411	0.9020	720
2	429	197	232	0.9001	718

Tb, Pb, and Nb refer to the number of total, positive, and negative correlation bands that passed the 0.01 significance test, respectively (400–2400 nm); rmax refers to the maximum absolute value of correlation coefficient.

**Table 3 plants-12-00501-t003:** Results of VCPA-GA hybrid strategy based on the hyperspectral datasets (0–2 order).

Orders	Nvar	Nlvs	RMSEC	RMSECV	RMSEP
0	5	9	3.41	3.61	5.74
0.2	54	10	3.35	3.59	5.42
0.4	16	9	3.38	3.58	5.54
0.6	18	10	3.33	3.55	5.41
0.8	14	10	3.16	3.33	5.04
1	27	8	3.05	3.34	5.61
1.2	19	8	2.81	3.04	5.51
1.4	11	9	2.82	3.14	5.24
1.6	32	9	2.59	3.12	5.86
1.8	15	5	2.82	3.15	5.25
2	43	5	2.59	3.02	5.39

Nvar and NIvs refer to the number of selected variables and the number of optimal PLS latent variables. RMSEC, RMSECV, and RMSEP refer to the root mean square error in calibration, the root mean square error in cross validation, and root mean square error in prediction.

**Table 4 plants-12-00501-t004:** Results of accuracy indicators for all MLRAs based on the hyperspectral datasets (0–2 order), the units of MAEs and RMSEs are$\;\mu g\cdot {\mathrm{cm}}^{-2}$.

Orders	Algorithm	Training Set			Testing Set			Validation Set			RPIQ
		R^2^	MAE	RMSE	R^2^	MAE	RMSE	R^2^	MAE	RMSE	
0	RR	0.82	3.56	4.61	0.82	4.08	5.10	0.83	3.94	5.04	2.77
	RF	0.85	3.31	4.30	0.66	5.45	6.94	0.66	6.02	7.20	1.67
	XGBoost	0.99	0.53	0.71	0.67	5.46	6.82	0.72	5.34	6.51	1.99
	SVR	0.85	3.14	4.26	0.84	3.72	4.76	0.84	3.93	4.92	3.07
	GPR	0.85	3.29	4.30	0.83	3.79	4.87	0.85	3.88	4.84	3.13
0.2	RR	0.86	3.17	4.06	0.84	3.75	4.71	0.88	3.40	4.23	3.59
	RF	0.91	2.51	3.29	0.77	4.30	5.74	0.80	4.28	5.47	2.57
	XGBoost	0.98	1.21	1.70	0.83	3.84	4.96	0.85	3.90	4.84	2.67
	SVR	0.88	2.69	4.74	0.84	3.51	4.74	0.81	3.94	5.37	3.09
	GPR	0.87	3.03	3.89	0.85	3.67	4.62	0.88	3.55	4.29	3.86
0.4	RR	0.84	3.27	4.43	0.84	3.71	4.83	0.82	4.11	5.21	2.88
	RF	0.90	2.64	3.48	0.73	4.74	6.17	0.71	5.46	6.60	1.99
	XGBoost	0.94	2.06	2.75	0.76	4.38	5.86	0.73	5.18	6.37	2.47
	SVR	0.86	2.95	4.15	0.82	3.82	5.08	0.21	5.40	10.92	1.29
	GPR	0.86	3.18	4.09	0.86	3.51	4.43	0.84	3.99	4.88	2.89
0.6	RR	0.87	3.06	4.01	0.85	3.72	4.68	0.86	3.75	4.65	3.44
	RF	0.90	2.69	3.54	0.81	2.69	3.54	0.84	3.91	4.95	3.08
	XGBoost	0.91	2.56	3.36	0.80	4.03	5.27	0.84	3.98	4.87	3.13
	SVR	0.86	2.87	4.06	0.84	2.53	4.75	0.83	3.96	5.14	3.42
	GPR	0.88	2.99	3.83	0.85	3.70	4.68	0.87	3.75	4.52	3.89
0.8	RR	0.86	3.10	4.04	0.84	3.81	4.74	0.86	3.76	4.68	3.38
	RF	0.91	2.55	3.27	0.81	4.17	5.20	0.85	3.78	4.84	2.81
	XGBoost	0.92	2.41	3.10	0.80	4.21	5.36	0.82	4.09	5.27	2.41
	SVR	0.86	3.03	4.11	0.82	3.91	5.05	0.86	3.55	4.58	3.36
	GPR	0.87	2.98	3.88	0.85	3.74	4.69	0.85	3.86	4.77	3.23
1	RR	0.89	3.02	3.71	0.85	3.68	4.67	0.86	3.60	4.63	3.34
	RF	0.89	2.78	3.60	0.84	3.63	4.72	0.85	3.79	4.75	2.48
	XGBoost	0.91	2.53	3.25	0.84	3.82	4.71	0.84	3.91	4.98	2.59
	SVR	0.84	3.23	4.45	0.82	3.95	5.04	0.83	3.90	5.06	3.05
	GPR	0.89	3.02	3.71	0.85	3.68	4.67	0.86	3.60	4.63	3.34
1.2	RR	0.87	3.10	4.00	0.85	3.77	4.63	0.83	4.03	5.10	2.86
	RF	0.90	2.58	3.49	0.83	3.73	4.91	0.83	3.78	5.10	2.50
	XGBoost	0.93	2.27	2.96	0.83	3.72	4.99	0.82	3.96	5.18	2.82
	SVR	0.88	2.58	3.84	0.85	3.61	4.68	0.84	4.06	4.96	2.94
	GPR	0.87	3.09	3.99	0.85	3.77	4.64	0.83	4.05	5.13	2.85
1.4	RR	0.84	3.32	4.38	0.84	3.72	4.79	0.82	4.15	5.20	2.99
	RF	0.88	2.90	3.78	0.81	3.76	5.19	0.81	4.24	5.37	2.62
	XGBoost	0.89	2.67	3.63	0.79	4.15	5.46	0.78	4.69	5.74	2.58
	SVR	0.86	2.72	4.09	0.81	4.09	5.24	0.82	4.27	5.29	2.88
	GPR	0.84	3.32	4.26	0.84	3.68	4.76	0.82	4.16	5.21	2.92
1.6	RR	0.86	3.11	4.07	0.85	3.74	4.59	0.86	3.56	4.68	2.98
	RF	0.92	2.42	3.19	0.81	3.82	5.24	0.85	3.52	4.75	2.79
	XGBoost	0.99	0.88	1.24	0.80	3.80	5.35	0.82	4.04	5.28	2.71
	SVR	0.86	2.94	4.16	0.84	3.70	4.77	0.83	3.96	5.09	2.85
	GPR	0.84	3.31	4.35	0.85	3.70	4.66	0.85	3.67	4.75	3.06
1.8	RR	0.84	3.41	4.37	0.85	3.59	4.67	0.84	3.78	4.96	2.89
	RF	0.91	2.47	3.25	0.81	3.76	5.14	0.86	3.42	4.63	2.80
	XGBoost	0.97	1.36	1.80	0.81	3.81	5.14	0.84	3.78	4.95	2.88
	SVR	0.83	3.27	4.47	0.83	3.74	4.95	0.81	3.97	5.30	2.70
	GPR	0.83	3.47	4.46	0.84	3.68	4.78	0.84	3.73	4.91	2.91
2.0	RR	0.86	3.21	4.09	0.83	3.93	4.87	0.82	3.93	4.87	2.66
	RF	0.87	3.06	3.89	0.82	3.83	5.10	0.84	3.71	4.94	2.67
	XGBoost	0.93	2.23	2.86	0.81	3.87	5.24	0.85	3.34	4.73	2.69
	SVR	0.87	2.60	3.88	0.81	3.99	5.22	0.77	4.59	5.93	2.26
	GPR	0.85	3.31	4.27	0.84	3.76	4.83	0.83	3.97	5.10	2.75

## Data Availability

Data available on request from the authors.

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
