# Peer review of "Retrieval of Leaf Chlorophyll Contents (LCCs) in Litchi Based on Fractional Order Derivatives and VCPA-GA-ML Algorithms"

_plants, 2023, doi:10.3390/plants12030501_

Round 1

Reviewer 1 Report

First of all I gonna mention that this paper is a good piece of work and may be published in Plants after minor revision.

However there are some drawbacks that maybe can be improved in the revised version, or in future works.

First of all, this concerns the choice of biological samples. The authors are absolutely correct saying that 'LCC is a key indicator of crop's physiological status, and changes in it can be used to assess a crop's photosynthetic ability, growth and development stage, nutrition, stress from humans or the environment, illnesses, and pests' (lines 215-217). However, they say nothing about physiological status of the trees used in this study. Did the selected trees have variations in it, or all of them were in good conditions, and the observed variations in LCC were in some normal range? If the former, this should be described in Methods section; otherwise the model should be re-evaluated on stressed plants in future work.

Use of fractal order derivatives to extract features from hyperspectral data seems to be a promising approach, however in this work I cannot see any convincing evidence for its advantage over traditional methods. Differences between the fractal order derivatives and traditional 1st and 2nd order derivatives in rmax (Table 3) are almost negligible. In Table 4 RMSEP shown the best result for order 0.8, but RMSECV (and RMSEC) - for order 2. As for me, I generally prefer cross validation metrics as they allow evaluating not only point estimate for the metric, but also its standard deviation. If possible, it would be valuable if std for RMSECV is also presented in Table 4.

In this work, LCC obtained by SPAD-502 plus portable chlorophyll meter measurements were used as ground truth. However, these SPAD provides rather rough LCC estimations based on in situ two-wavelength measurements. To get more reliable ground truth data the authors should perform chlorophyll extraction and measure chlorophyll content in extracts.

Minor issues:

Please ensure that acronyms for all metrics used are spelled out at first use.

page 1 line 21 - preposition missing

page 1 line 37 - superfluous word 'efficient'

page 2 line 46 - extra comma

page 2 line 83 - Its -> its

page 7 line 64 - what is É·?

Author Response

Dear reviewer,
Thank you for the comments! We benefit a lot from these valuable comments. We've revised the manuscript. Please check the details in document. 

Best,

Dan Li

Reviewer 2 Report

In this study, the Authors present a proposal to use machine learning regression algorithms using a set of features selected with different approaches from hyperspectral data. The work is relevant and the results are very promising. However, there are remarks whose explanation in the text or recalculation will allow for a credible and unambiguous assessment of the proposed solution. The structure of the work is good, but the description needs to be detailed.

General Notes to the Manuscript:

1. Enter the abbreviations the first time you use the term (also applies to abstract: LCC?).

2. Quality of the drawings: the resolution needs to be improved, they are very blurry. It is almost impossible to read the values and text.

3. Systematize the spaces between the value and the unit.

4. Article template assumes the names Figure, Table -- this form should be used in the text (not Fig.)

5. There are many statements in the text, but they do not refer to literature. The text should be verified and much more anchored in the current state of knowledge -- apart from the introduction Authors rarely use references.

Details:

1. “LCC was significantly increased, from 0.8542 to 0.9179”

  What test was the materiality assessment carried out? What is the p-value?

2. Table 1:

- it is worth specifying columns from the Type of spectra or image data column: ‘data source’ (satallite or airborne) and ‘spectral type’ (HSI or MSI) and simply the name of the ‘sensor’ (e.g. Landsat-8). It would be good to add a column with the year of references, this will allow for a cross-sectional presentation of how methods change over time.

  - not all abbreviations in Table 1 have expansion (eg SVM).

3. “Hyperspectral data for the Litchi leaves were measured by ASD FieldSpec3 spectrometer (Analytical Spectral Devices, Inc., Boulder, CO, USA) with the range of 350-2500 nm. “

Reference to documentation

4. “The edge bands 350-399 nm and 2401-2500 nm with high optical noise were removed.”

Is this a literature-defined range that should be removed?

5. Savitzky-Golay filtering method

Why was this method used? Literature reference.

6. “Then the fractional order derivative (FOD) of the smoothed spectral data were calculated by Grünwald-Letnikov (G-L) algorithm as shown in the Equation (1) using a program in Matlab R2021a (The MathWorks Inc.: Natick, MA , USA).”

Reference to literature and documentation, especially for formula (1).

7. How many images were collected, 490?

8. It is not clear at what level of data the cross-validation was performed. If it's litchi tree level then it's fine (split based on tree IDs). If it was carried out at the sample level, there is an information leak between the training and test sets, which may result in overly optimistic results when evaluating the data. At what level was the data splitted?

9. Was the standardization of the data carried out before the splitting into folds? If so, there is also a risk of information leakage. The data should be standardized within the training set and based on the mean and sd from this set, the test and validation set should be processed (for a given experiment, i.e. it should be repeated as many times as there are folds).

10. What is the source of Figure 4? References to the literature from which the characteristic ranges of the spectral curve are inferred. I generally observe a lack of literature references for many sentences, for example “Chlorophyll, which has a strong absorption of blue and red light and a high reflection of green light, is primarily responsible for this property.”

11. It is good to show the exact p-value for the significance test. Why was the threshold of 0.01 adopted and not the classic 0.05?

12. After selecting the best sensitive band combination - it's not clear to me if the feature selection was before or after the division into folds? If before then the next defeat is a high risk of information leakage. Make a selection for the training sets and see which of the features are repeated in each of them.

Author Response

Dear reviewer,
Thank you for the helpful comments. We read the comments  and made the responses carefully. Please check the details from the document. 

Best,
Dan Li

Round 2

Reviewer 2 Report

Thank you very much for implementing my comments and congratulations on a very interesting article.